# Oncological Outcomes of Distal Ureterectomy for High-Risk Urothelial Carcinoma: A Multicenter Study by The French Bladder Cancer Committee

**DOI:** 10.3390/cancers14215452

**Published:** 2022-11-06

**Authors:** Alexandra Masson-Lecomte, Victoire Vaillant, Mathieu Roumiguié, Stéphan Lévy, Benjamin Pradère, Michaël Peyromaure, Igor Duquesne, Alexandre De La Taille, Cédric Lebâcle, Adrien Panis, Olivier Traxer, Priscilla Leon, Maud Hulin, Evanguelos Xylinas, François Audenet, Thomas Seisen, Yohann Loriot, Yves Allory, Morgan Rouprêt, Yann Neuzillet

**Affiliations:** 1Department of Urology, APHP, Hôpital Saint Louis, 75010 Paris, France; 2Inserm, CEA, HIPI, Université Paris Cité, 75475 Paris, France; 3Department of Urology, APHP, Hôpital Henri Mondor, Université Paris-Est Créteil, 94000 Créteil, France; 4Department of Urology, CHU de Toulouse, UPS, Université de Toulouse, 31000 Toulouse, France; 5Department of Urology-UROSUD, La Croix du Sud Hospital, 31130 Quint-Fonsegrives, France; 6Department of Urology, Comprehensive Cancer Center, Medical University of Vienna, 1090 Vienna, Austria; 7Department of Urology, APHP, Hôpital Cochin, Université Paris Cité, 75014 Paris, France; 8Department of Urology, APHP, Hôpital Bicêtre, Université Paris Sud-Saclay, 94270 Le Kremlin-Bicêtre, France; 9GRC 10 Lithiase Urinaire, Department of Urology, APHP, Tenon Hospital, Sorbonne University, 75020 Paris, France; 10Department of Urology, clinique Pasteur, 17200 Royan, France; 11Department of Urology, APHP, Hôpital Bichat Claude-Bernard, Université Paris Cité, 75014 Paris, France; 12Department of Urology, APHP, Hôpital Européen Georges Pompidou, Université Paris Cité, 75014 Paris, France; 13GRC 5 Predictive Onco-Urology, Department of Urology, APHP, Pitié Salpêtrière Hospital, Sorbonne University, 75013 Paris, France; 14Department of Cancer Medicine, INSERM U981, Gustave Roussy, Université Paris-Saclay, 94805 Villejuif, France; 15Department of Pathology, Curie Institute, University of Paris-Saclay–UVSQ, 92210 Saint-Cloud, France; 16Department of Urology, Hôpital Foch, University of Paris-Saclay–UVSQ, 92150 Suresnes, France

**Keywords:** distal ureterectomy, urothelial carcinoma, cancer, high risk, cancer-specific survival, intravesical recurrence-free survival

## Abstract

**Simple Summary:**

Radical nephroureterectomy (RNU) is the standard treatment for high-risk upper tract urothelial carcinoma (UTUC). It implies significant reduction in the renal function, compromising adjuvant chemotherapy administration and leading to risk of end stage renal disease in frail patients. Distal ureterectomy (DU) might be an alternative for tumors of the distal ureter but its indications remain unclear mainly due to concern about potential upper tract recurrences. The objective of our retrospective study was to determine the oncologic outcomes of DU for high-risk UTUC of the pelvic ureter, and to assess factors associated with recurrence in the particular population. We showed that oncological outcomes after DU were similar to those after RNU. Some of the classical prognostic factors after RNU were not identified in this cohort, pinpointing the necessity to consider tumors of the distal ureter as a specific entity.

**Abstract:**

Upper urinary tract urothelial carcinoma (UTUC) is an uncommon disease and its gold-standard treatment is radical nephroureterectomy (RNU). Distal ureterectomy (DU) might be an alternative for tumors of the distal ureter but its indications remain unclear. Here, we aimed to evaluate the oncological outcomes of DU for UTUC of the pelvic ureter. We performed a multicenter retrospective analysis of patients with UTUC who underwent DU. The primary endpoint was 5-year cancer-specific survival (CSS), followed by overall survival (OS), intravesical recurrence-free (IVR) and homolateral urinary tract recurrence-free (HUR) survivals as secondary endpoints. Univariate and multivariate Cox regressions were performed to assess factors associated with outcomes. 155 patients were included, 91% of which were high-risk. 5-year CSS was 84.4%, OS was 71.9%, IVR-free survival was 43.6% and HUR-free survival was 74.4%. Multifocality, high grade and tumor size were the most significant predictors of survival endpoints. Of note, neither hydronephrosis nor pre-operative diagnostic ureteroscopy/JJ stent were associated with any of the endpoints. Perioperative morbidity was minimal. In conclusion, DU stands as a possible alternative to RNU for UTUC of the pelvic ureter. Close monitoring is mandatory due to the high risk of recurrence in the remaining urinary tract.

## 1. Introduction

Upper urinary tract urothelial carcinoma (UTUC) is a rare disease, representing 5 to 10% of urothelial carcinomas [1]. This malignancy can arise from both the renal pelvicalyceal system or the ureter, the latter occurring in most cases in its distal part [2]. According to the EAU guidelines, UTUC patients should be risk stratified to select those with low-risk disease (i.e., single, <2 cm, low grade, unifocal) for kidney sparing surgeries, including endoscopic management or ureterectomy [3].

As such, radical nephroureterectomy (RNU) with bladder cuff excision currently represents the standard of care only for patients with high-risk UTUC, regardless of tumor location [4]. However, this procedure is associated with significant post operative decrease in renal function [5], potentially compromising the administration of adjuvant cisplatin-based chemotherapy for patients with advanced disease as supported by the POUT trial [6].

Interestingly, distal ureterectomy (DU) stands on the edge of both kidney-sparing surgery and radical excision, given that it allows for complete removal of the compromised ureter while preserving the kidney unit. Concomitant regional lymph node dissection can also be performed at the time of DU, ensuring proper pelvic staging. However, the latest EAU guidelines restricted the use of DU as an alternative to RNU only in high-risk tumors with imperative indications, such as solitary kidney, bilateral tumors or pre-existing renal failure [7,8]. Indeed, the role of DU for treating UTUC of the distal ureter in elective indications remains unclear, based on the limited evidence available in the literature. In addition, it is noteworthy that the risk of recurrence in the remaining urothelial tract after DU has been scarcely evaluated to date.

Against this backdrop, we aimed to evaluate oncological outcomes after DU for UTUC in a large multicenter cohort of patients treated for both elective and imperative indications.

## 2. Material & Methods

### 2.1. Patients’ Selection (Figure 1: Flow Chart)

We retrospectively identified all DUs performed between January 2010 and December 2020, among 12 urology departments through our collaborative network from the Bladder Cancer Committee of the French Association of Urology. Procedures were searched using the French CCAM (Classification Commune des Actes Médicaux) codes JCFA003 (DU with ureteral anastomosis), JCFA008 (DU with ureterovesical reimplantation and anti-reflux assembly), JCFA009 (DU with ureterovesical reimplantation alone) and JCFA010 (DU with ureterovesical reimplantation and a psoas-bladder hitch).

The main inclusion criterion was the presence of UTUC on the final pathology specimen. Patients who received DU for benign conditions were not included (endometriosis, ureteral strictures, stones etc.). Exclusion criteria included history or concomitant bladder or upper tract tumor infiltrating the muscle, the identification of distant metastases at initial diagnosis, missing data for the EAU risk classification and lack of follow-up.

**Figure 1 cancers-14-05452-f001:**
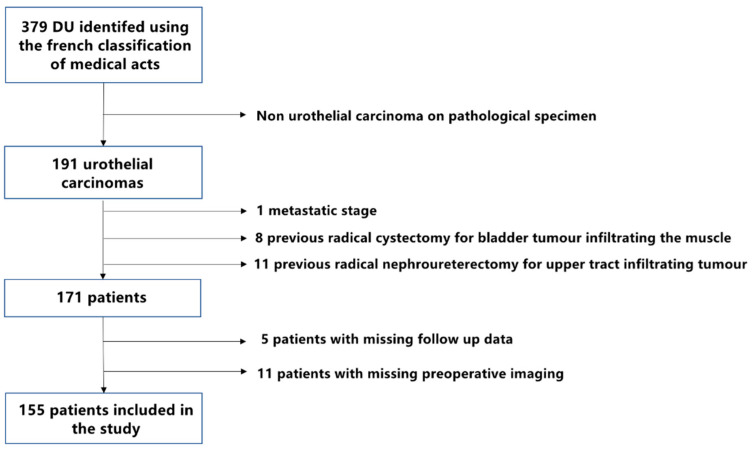
Flow chart for patients’ selection.

### 2.2. Variables Retrieval and Definition

#### 2.2.1. Patient Characteristics

For all included patients, we collected age at surgery, gender, Charlson comorbidity index [9], body mass index, immediate pre-operative estimated glomerular filtration rate (eGFR), prior or synchronous non-muscle-invasive bladder cancer, abdominal computed-tomography (CT) report (which described local invasion, hydronephrosis and enlarged pelvic nodes), and information on the use of preoperative JJ stent. Preoperative renal failure was defined as an eGFR < 60 mL/min.

#### 2.2.2. Surgical Technique

DU was performed using an open or laparoscopic approach (standard or robot assisted). All patients had a complete removal of the distal ureter from the iliac vessels to the bladder cuff, with intraoperative ureteral frozen section analysis in most cases. Bladder cuff excision was performed using either an extravesical, transvesical or endoscopic approach. Lymphadenectomy was performed at the surgeons’ discretion.

#### 2.2.3. Perioperative Variables

The type of surgical approach (open or mini-invasive, robot-assisted or not), operative time, blood loss, use of lymphadenectomy, distal ureter management (extravesical, transvesical or endoscopic), length of stay (defined as days from surgery until discharge), and Clavien classification for 30-day postoperative complications [10] were retrieved from patient charts.

#### 2.2.4. Pathological Variables

Finally, we collected pathological stage (according to the 2017 Union for International Cancer Control classification), tumor grade (according to the 1973 and 2004 World Health Organization grading system), as well as information on tumor focality, size (defined on pathological specimen), location (defined on imaging or ureteroscopy if performed), and 2022 EAU risk group. High-risk UTUC was defined by the presence of one from any of the following criteria: multifocality, tumor size ≥ 2 cm, high-grade cytology or biopsy, local invasion on CT, hydronephrosis, previous radical cystectomy for high-grade bladder cancer, variant histology. Low-risk UTUC was defined by the absence of all the aforementioned criteria. In addition, we retrieved pN stage and the total number of lymph nodes removed as well as surgical margin status. The distal margins were those above and below the surgical resection while lateral margins were those in the periureteral fat circumferentially to the ureter.

### 2.3. Endpoints

Our primary endpoint of the study was the 5-year cancer-specific survival (CSS).

Secondary oncological endpoints were 5-year overall survival (OS), 5-year intravesical recurrence-free survival (IVR) defined as the occurrence of any pathologically proven non-muscle- or muscle-invasive urothelial carcinoma in the bladder, and 5-year homolateral upper tract recurrence-free survival (HUR) defined as the occurrence of any pathologically proven or typical on imaging UTUC in the remaining upper urinary tract.

Secondary perioperative endpoints were length of stay and the risk of 30-day postoperative complications based on the Clavien–Dindo classification.

### 2.4. Statistical Analysis

Continuous variables were described using mean (standard deviation) and median (min-max). Categorical variables were described using number and percentages. Survivals were modelized using the Kaplan–Meier method and compared using the Log rank test. To assess variables associated with survival endpoints, uni and multi variable Cox regression analyses were performed. *p* values < 0.05 were deemed significant. To ensure sufficient power in the multivariable analyses, a 5 to 10 events per variable cut off was applied. In cases where all the significant variables in the univariate analyses could not be included, factors used to define the EAU risk classification were prioritized. Statistical analyses were performed using RStudio 1.2.5042, © 2009–2020 RStudio, Inc. (Boston, MA, USA).

## 3. Results

### 3.1. Patients’ Characteristics

Overall, 155 UTUC patients treated with DU were included in the present study. Patients’ characteristics are summarized in Table 1. Based on the 2022 EAU risk classification, 91% of them had high-risk disease including 42 cases with high-grade tumors based on either biopsy/cytology, 84 cases with tumor size ≥ 2 cm, 8 cases with clinical evidence of lymph nodes invasion, and 114 cases with hydronephrosis. Eighteen patients (12%) were classified as having high-risk UTUC solely based on the presence of preoperative hydronephrosis.

In most cases, DU was performed in elective indications. Fifty-five (35.5%) patients had a moderate to severe renal failure before surgery (eGFR < 60 mL/min), three patients had a previous pTa UTUC and one patient had a concomitant contralateral pTaG1 UTUC treated with RNU.

Preoperative biopsy was performed in 104 (67.1%) patients, which was contributive for stage and grade in 86.5% and 88.5% of cases, respectively. Other patients had highly evocative imaging alone or with either positive cytology or history of bladder cancer or synchronous bladder cancer.

A preoperative JJ stent was inserted in 74 (47.7%) patients.

### 3.2. Operative and Pathological Characteristics (Table 2)

An open approach was used to perform DU in 141 (91%) patients and the bladder-cuff removal was managed using the extravesical technique in 103 (66%) patients. A concomitant lymphadenectomy was performed in 55 (35.5%) patients.

Regarding the postoperative characteristics, the mean length of stay was 8.7 days after surgery. Fifty-two (33.4%) patients had a postoperative complication, including ten (6.4%) patients with Clavien–Dindo grade III who required open surgery (laparotomy for acute bleeding or evisceration), endoscopic (JJ stent or nephrostomy) or radiological treatment (urine fistulae requiring percutaneous drainage). Clavien–Dindo grade IV complications occurred in 3 (1.9%) patients (one intestinal obstruction caused by early post operative adhesions, one ischemic stroke and one acute obstructive pyelonephritis leading to septic shock) and one patient died (Mendelson syndrome secondary to severe ileus, Grade V).

With regard to final pathology, 112 (72.2%) patients had non-muscle invasive UTUC. High-grade features were observed in 102 (65.8%) patients. Among patients who received lymph node dissection, 3 (1.8%) had lymph node invasion. Overall, 34 (22%) patients had positive distal or lateral margins.

**Table 2 cancers-14-05452-t002:** Operative and pathological characteristics.

Total	155
Surgical approach - Open - Mini-invasive - Missing	141 (91%) 11 (9%) 3 (1.9%)
Operative time Mean (SD)	165.6 (58.9)
Blood loss (mL) Mean (SD)	268.1 (246.4)
Lymphadenectomy	55 (35.4%)
Distal ureter management - Extravesical - Transvesical - Endoscopic - Missing	103 (66.4%) 37 (23.8%) 7 (4.6%) 8 (5.2%)
LOS Mean (SD)	8.7 (7.6)
Post-operative complications - Clavien 1 - Clavien 2 - Clavien 3a/3b - Clavien 4 - Clavien 5	14 (9%) 24 (15.5%) 1/9 (0.6%/5.8%) 3 (1.9%) 1 (0.6%)
1973 DU grade - G1 - G2 - G3 - Missing	16 (10.3%) 39 (25.1%) 90 (58%) 10 (6.4%)
2004 DU grade - LG - HG - Missing	44 (28.3%) 102 (65.8%) 9 (5.8%)
pT - pT0 - pTa - pT1 - pTis - pT2 - pT3 - pT4	6 (3.9%) 73 (47.1%) 32 (20.7%) 1 (0.6%) 20 (12.9%) 20 (12.9%) 3 (1.9%)
pN - pN0 - pN1 - pN2 - pNx	49 (31.7%) 1 (0.6%) 2 (1.3%) 103 (66.4%)
Total Nodes Mean (SD)	5.8 (5.5)
Margins - Distal positive margins - Lateral positive margins	23 (14.8%) 11 (7%)

### 3.3. Oncological Outcomes

#### 3.3.1. Overall Survival

Kaplan–Meier analysis showed that 5-year OS was 71.9% (Figure 2). Results from Cox regression analyses for factors associated with OS are presented in Table 3. In univariable analysis, age (=0.01), Charlson score (*p* = 0.07), cT2 stage (*p* = 0.08), cN + stage (*p* = 0.002), multifocality (*p* = 0.01), high-grade biopsy (*p* = 0.08) and tumor size (*p* = 0.03) were significantly associated with OS. In multivariable analysis, multifocality (HR = 4.98; 95%CI = 0.80–30.93; *p* = 0.08), high-grade biopsy (HR = 3.46; 95%CI = 1.01–11.96; *p*= 0.04) and tumor size (HR = 1.07; 95%CI = 1.02–1.12; *p* = 0.004) were independent predictors of OS.

#### 3.3.2. Cancer-Specific Survival

Kaplan–Meier analysis showed that 5-year CSS was 84.4% (Figure 3). Results from Cox regression analyses for factors associated with CSS are presented in Table 4. In univariable analysis, tumor size (*p* = 0.01), cT2 stage (*p* = 0.025), cN + stage (*p* = 0.007), multifocality (*p* = 0.06), high-grade biopsy (*p* = 0.07), high-grade cytology (*p* = 0.09) and positive margins (distal or lateral) (*p* < 0.01) were significantly associated with CSS. Hydronephrosis and prior/synchronous NMIBC were not. In multivariable analysis, tumor size (HR = 1.09; 95%CI = 1.02–1.15; *p* = 0.004) and high-grade biopsy (HR = 8.77; 95%CI = 0.96–79.53; *p* = 0.05) were independent predictors of CSS.

#### 3.3.3. Intra Vesical Recurrence

Kaplan–Meier analysis showed that 2- and 5-year IVR-free survival were 58% and 43.6%, respectively (Figure 4). A total of 67 patients experienced IV recurrence during follow up (43.2%): 14 (20.9%) Ta G1/LG, 16 (23.9%) Ta G2, 13 (19.4%) Ta G3/HG, 12 (17.9%) T1G3/HG, 3 (4.5%) isolated CIS, 3 (4.5%) MIBC and 6 (8.9%) patients had missing data for pathology of IV recurrence.

The results from Cox regression analyses for factors associated with IVR are presented in Table 5. In univariable analysis, tumor size (*p* = 0.04), cT2 stage (*p* = 0.09), and distal positive margins (*p* = 0.018) were significantly associated with IV recurrence. Past/present smoking, prior/synchronous NMIBC, hydronephrosis, high grade cytology, preoperative JJ stent, diagnostic URS and distal ureter management were not associated with IV recurrence. In multivariable analysis, tumor size (HR = 1.01; 95%CI = 0.99–1.03; *p* = 0.06) and distal positive margins (HR = 1.93; 95%CI = 1.01–3.70; *p* = 0.04) were independent predictors of IVR.

#### 3.3.4. Homolateral Upper Tract Recurrence

Kaplan–Meier analysis showed that 2- and 5-year HUR-free survival were 83.1% and 74.4%, respectively (Figure 5). A total of 28 patients experienced homolateral upper tract recurrences during follow up (18%). A total of 12 HUR were G3/HG (43%) and 3 were stage T2 and more. Overall, 14 HUR were treated with RNU, 3 with RNU and radical cystectomy, 1 was managed endoscopically, 3 had only systemic chemotherapy because of concomitant metastatic progression, and 3 were only watched with no radical treatment due to imperative indications. Data about management of HUR were missing for four patients. The results from Cox regression analyses for factors associated with HUR are presented in Table 6. In univariable analysis, tumor size (*p* = 0.02) and multifocality (*p* = 0.07) were significantly associated with HUR. Hydronephrosis, high-grade biopsy, preoperative JJ stent, diagnostic URS, prior/synchronous NMIBC, and distal positive margins were not associated with HUR. In multivariable analysis, tumor size (HR = 3.59; 95%CI = 1.22–10.60; *p* = 0.02) and multifocality (HR = 3.44; 95%CI = 1.01–11.71; *p* = 0.04) were independent predictors of HUR.

## 4. Discussion

Available evidence is limited regarding oncological effectiveness of DU for high-risk or even high-grade UTUC. Established prognostic factors for UTUC have been published using series of RNU and may not be applicable to patients treated with DU. We report here one of the largest series of DU performed at academic hospitals in a contemporary period.

Most patients treated with DU in our cohort had high-risk UTUC. The retrospective nature of the study makes it difficult to properly dichotomize patients between elective and imperative DU, especially as it might be a subjective matter. However, almost 40% of them had obvious imperative indications. This pinpoints that the EAU guidelines reserving DU for imperative situations are poorly followed in daily practice.

With regard to patients’ characteristics, the median age was 72 years old with a median Charlson score of 6 which is similar to populations of patients treated with RNU [4]. However, final pathology was different from what has been observed in RNU cohorts, given that most patients (72.2%) had non-muscle-invasive UTUC, with 47% of pTa. In a retrospective study by Fang et al. comparing oncological outcomes in RNU and DU for urothelial carcinoma of the middle or distal ureter, the rate of non-muscle-invasive diseases (pTa or pT1) was 52.9% and 89.2% after RNU and DU, respectively [11]. Our study population reflects the reluctance of care givers to remove the entire upper urinary tract in selected patients with high-risk UTUC located in the distal part of the ureter. Nonetheless, high-risk UTUC is a heterogenous group of tumors and it can be hypothesized that we report here a selected cohort of DU for high risk. New risk stratification models are being developed [12,13,14], in line with this observation, helping in the future to better refine treatment indications in high-risk tumors.

When it comes to the surgical technique, DU is associated with low morbidity (minimal bleeding, relatively short operating time). Lymphadenectomy can be performed in the same conditions than during RNU if indicated. In our experience, the 30-day postoperative complications rate was low, with only 6.4% of patients who needed a reintervention, and 2.5% of life-threatening complications. The relatively long length of stay was mainly related to the usual 7-day delay before removal of the urinary catheter.

The main advantage of the DU remains the kidney preservation. It has been proven that after RNU the renal function is lowered by about 20% [5,15] with a significant decrease in eGFR of 9.32 mL/min/1.73 m [2] in comparison with DU [16]. Chronic kidney disease reduces eligibility to adjuvant cisplatin-based chemotherapy, and increases the risk of cardiovascular morbidity, both with potential consequences on OS [17].

With regard to survival, CSS and OS after DU were, respectively 84.4% and 71.9% which are comparable to survival outcomes after RNU. In a multicenter international study including 2681 patients, Xylinas et al. showed that CSS after RNU ranged from 70% to 82% depending on the distal ureteral management [4]. Studies comparing oncological outcomes of RNU and DU have been published. A systematic review of 11 retrospective studies with 3963 patients comparing RNU and DU demonstrated no significative difference in CSS, OS and IVR between the two techniques [16]. Similarly, Jeldres et al. observed no difference in CSS between RNU and DU (82.2% and 86.6%, respectively) [18].

Tumor size, biopsy grade and multifocality were associated with CSS and OS, which is in line with many studies [19,20,21]. However, hydronephrosis was not associated with any of the oncological outcomes. Hydronephrosis is one of the high-risk features for UTUC according to the EAU risk stratification. In several RNU studies, hydronephrosis was a strong predictor of adverse pathological and oncological outcomes [22,23,24]. In addition, hydronephrosis has also been shown to be associated with IVR [11]. However, our results suggest that the presence of hydronephrosis had no adverse impact on DU outcomes and may preclude the use of DU. Of note, it is interesting to pinpoint that in our cohort, grade was not associated with IVR or upper tract recurrence but only OS and CSS endpoints. From our point of view, this supports the use of DU in high-grade patients as it is unlikely that removing the entire renal unit will impact cancer-specific survival. This is of particular importance in patients with existing chronic kidney disease, solitary kidney or comorbidities (imperative indications for renal preservation) as RNU would have life-threatening consequences. Alternatives to RNU are probably acceptable for those patients, in selected cases (add PMID 35475152 to reference list).

The 5-year IVR and HUR rates were as high as 56.4% and 25.6%, respectively. This may be related to the high rate of positive surgical margins (22%) which reflects the technical difficulties of managing UTUC in the distal ureter. In our study, the main factors associated with IVR were tumor size and positive distal margins while those associated with HUR were tumor size and multifocality. Interestingly, prior/synchronous NMIBC, preoperative JJ stent, diagnostic URS and distal ureter management were not associated with either IVR or HUR. Predictive factors of IVR have been evaluated after RNU. History of bladder cancer was associated with IVR in several studies [4,11,21,25], as well as the presence of CIS [26,27,28]. Endoscopic bladder cuff excision has also been shown to increase the risk of IVR [4]. Sharma et al. showed that the use of ureterorenoscopy with biopsy before RNU was strongly associated with intravesical recurrence [29]. Regarding HUR, Sountoulides et al. found that JJ stenting in patients with bladder cancer before cystectomy was a predictive factor of homolateral urinary tract recurrence by abolishing the anti-reflux mechanism of the intramural ureter, and causing a tumoral dissemination [30]. Our different results pinpoint that prognostic factors may be different after DU as compared to RNU. Nonetheless, very close monitoring after distal DU is mandatory, with imaging and ureterorenoscopy. The selection of cooperative patients before this kidney sparing surgery is imperative. Moreover, an extensive evaluation of the renal pelvis and calix should be performed before DU using direct visualization if possible and in situ cytology.

Our results should be interpreted within the limitations mainly related to its retrospective design and small sample size. The definition of imperative or elective indications was subjective, particularly in a population of elderly patients. There were data also lost to follow-up, and exclusions due to lack of imaging data, which was necessary for preoperative risk stratification. Proper risk group definitions would have needed a central review of preoperative CT scans as well as pathology specimens. Finally, the surgical technique for DU was not standardized, with different approaches and obvious bias between centers and surgeons.

## 5. Conclusions

The indications for DU to treat high-risk UTUC are unclear. In our experience, DU was minimally morbid, with OS and CSS comparable to RNU outcomes reported in the literature. Nevertheless, the high rates of IVR and HUR imply performing close monitoring. Prognostic factors were partly different from those observed after RNU, leading to refining patients’ selection. New risk categories may be needed depending on the type of treatment offered.

## Figures and Tables

**Figure 2 cancers-14-05452-f002:**
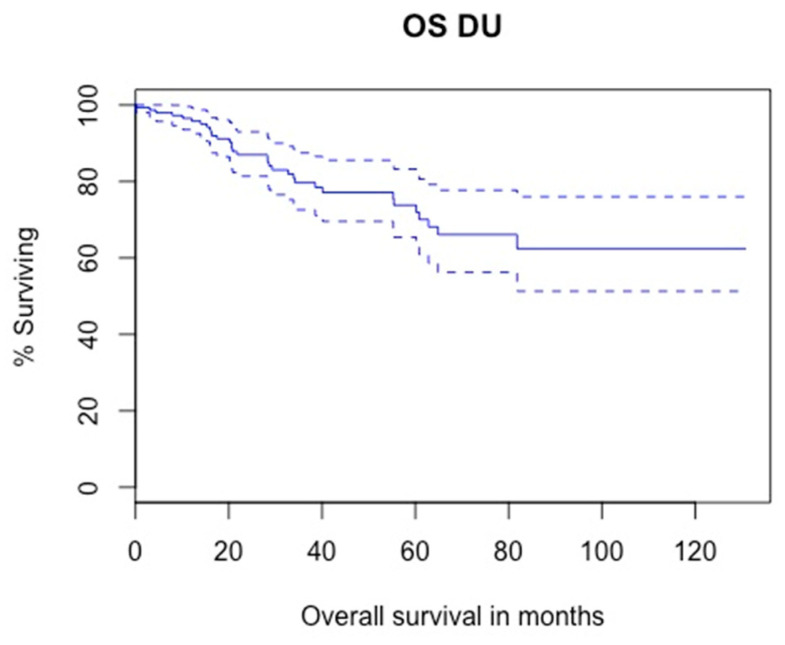
Kaplan–Meier modelling of overall survival (5-year OS: 71.9%, lower—upper 95%CI 0.632–0.819).

**Figure 3 cancers-14-05452-f003:**
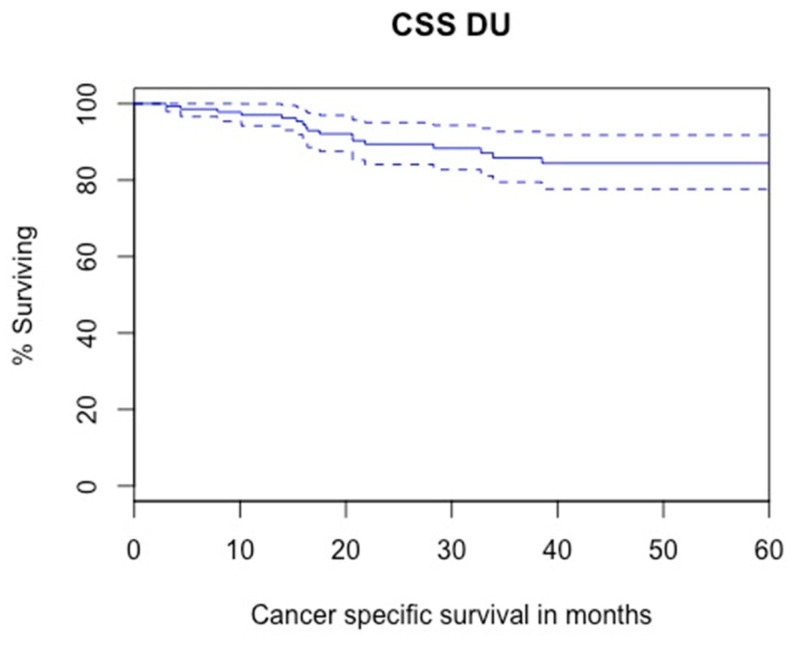
Kaplan–Meier modelling of cancer-specific survival (5-year CSS: 84.4%, lower—upper 95% CI 0.777–0.918).

**Figure 4 cancers-14-05452-f004:**
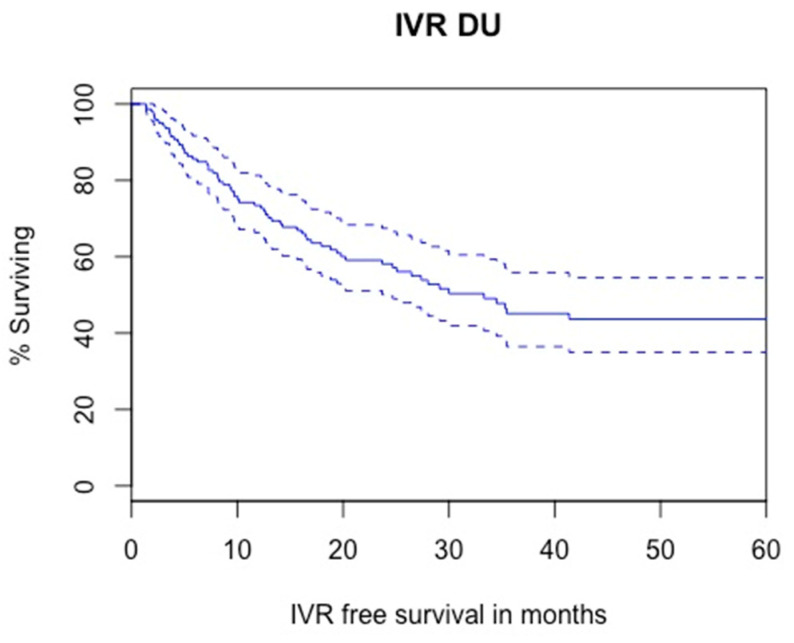
Kaplan–Meier modelling of intra vesical recurrence (5-year IV free survival: 43.6%, lower—upper 95% CI 0.349–0.545).

**Figure 5 cancers-14-05452-f005:**
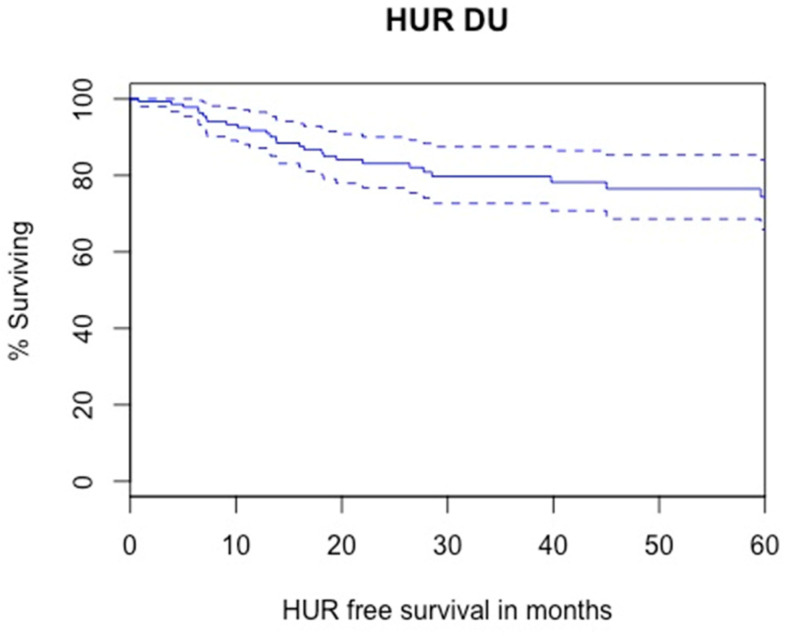
Kaplan–Meier modelling of homolateral upper tact recurrence (5-year HUR free survival: 74.4%, lower—upper 95% CI 0.658–0.840).

**Table 1 cancers-14-05452-t001:** Patients’ characteristics.

Total	155
AgeMean (SD)	72.6 (10.5)
Gender - Male Female	113 (72.9 %) 42 (27.1 %)
Charlson score Mean (SD)	6.19 (2.4)
BMI Mean (SD)	25.47 (4.9)
Pre-operative GFR - <60 mL/min - > or = 60 mL/lmin Missing	55 (35.5%) 66 (42.5%) 34 (22%)
Prior BC Synchronous BC	43 (27.7%) 28 (18%)
Abdominal CT report - Local invasion (>T2) - Hydronephrosis Enlarged pelvic nodes	5 (3.2%) 114 (73.5%) 8 (5.1%)
1973 Biopsy grade - G1 - G2 - G3 Missing	22 (21.2%) 35 (33.6%) 34 (32.7%) 13 (12.5%)
2004 Biopsy grade - LG - HG Missing	51 (49%) 41 (39.4%) 12 (11.6%)
cTNM - cTxN0M0 - cTaN0M0 - cTaN1M0 - cT1N0M0 - cT2N0M0 - cTxN1M0 - cTxN2M0 cT2N2M0	60 (38.7%) 69 (44.6%) 2 (1.3%) 16 (10.3%) 4 (2.6%) 2 (1.3%) 1 (0.6%) 1 (0.6%)
Multifocality	7 (4.5%)
Size (in mm) Mean (SD)	25.76 (14.1)
Location - Meatus - Distal ureter - Meatus + distal ureter Missing	7 (4.5%) 96 (61.9%) 39 (25.1%) 13 (8.3%)
Preoperative JJ stent	74 (47.7%)
EAU risk group Low High	14 (9%) 141 (91%)

**Table 3 cancers-14-05452-t003:** Cox regression for factors associated with OS.

	Univariate			Multivariate		
Variable	*p*	HR	95% CI	*p*	HR	95% CI
Age (continuous)	0.01 *	1.05	1.01–1.09	0.15	1.06	0.97–1.16
Gender Male Female	0.70	1.15	0.53–2.59			
Charlson (continuous)	0.07	1.14	0.98–1.33			
GFR >60 mL/min <60 mL/min	Ref 0.01 *	2.67	1.21–5.87	0.39	0.55	0.14–2.17
cTN cTaN0 cT1N0 cT2N0 cTanyN1–2 cTxN0M0	Ref 0.62 0.08 0.002 * 0.19	1.37 6.20 7.80 1.67	0.38–4.86 0.76–50.41 2.08–29.13 0.77–3.62			
Multifocality No Yes	Ref 0.01 *	3.95	1.38–11.42	0.08	4.98	0.80–30.93
2004 biopsy grade LG HG	Ref 0.08	2.29	0.88–5.92	0.04 *	3.46	1.01–11.96
Hydronephrosis No Yes	Ref 0.70	0.85	0.38–1.91			
Size (continuous)	0.03 *	1.02	1.00–1.04	0.004 *	1.07	1.02–1.12

* <0.05.

**Table 4 cancers-14-05452-t004:** Cox regression for factors associated with CSS.

	Univariate			Multivariate		
Variable	*p*	HR	95% CI	*p*	HR	95% CI
Age (continuous)	0.09	1.04	0.99–1.10			
Hydronephrosis No yes	Ref 0.99	1.00	0.32–3.08			
Size (continuous)	0.01 *	1.03	1.00–1.05	0.004 *	1.09	1.02–1.15
cTN cTaN0 cT1N0 cT2N0 cTanyN1–2 cTxN0M0	Ref 0.69 0.025 * 0.007 * 0.03 *	1.58 13.96 12.23 4.15	0.16–15.19 1.39–140.20 1.98–75.35 1.14–15.10			
Multifocality No Yes	Ref 0.06	4.02	0.90–17.91	0.12	6.05	0.60–61.05
2004 biopsy grade LG HG	Ref 0.07	7.25	0.84–62.14	0.05 *	8.77	0.96–79.53
Cytology Neg Atypical Positive for HG No Diag	Ref 0.86 0.09 0.20	0.81 4.00 2.99	0.07–9.02 0.77–20.69 0.54–16.40			
Past history NMIBC No Yes	Ref 0.55	0.71	0.23–2.18			
Synchronous NMIBC No Yes	Ref 0.99	1.00	0.28–3.50			
Distal margins Negative Positive	Ref <0.01 *	5.20	1.95–13.8			
Lateral Margins Negative Positive	Ref <0.01 *	13.72	5.02–37.46			

* <0.05.

**Table 5 cancers-14-05452-t005:** Cox regression for factors associated with IVR.

	Univariate			Multivariate		
Variable	*p*	HR	95% CI	*p*	HR	95% CI
Gender Male Female	0.8	0.92	0.52–1.65			
GFR >60 mL/min <60 mL/min	Ref 0.31	1.31	0.76–2.28			
Tobacco use Never Present Past	Ref 0.78 0.15	0.91 1.53	0.46–1.79 0.85–2.74			
Past history NMIBC No Yes	Ref 0.57	0.85	0.50–1.45			
Synchronous NMIBC No Yes	Ref 0.67	1.13	0.61–2.08			
Hydronephrosis No yes	Ref 0.95	1.01	0.57–1.78			
Size (continuous)	0.04 *	1.01	1.00–1.03	0.06	1.01	0.99–1.03
cTN cTaN0 cT1N0 cT2N0 cTanyN1–2 cTxN0M0	Ref 0.33 0.09 * 0.23 0.07 *	1.50 3.47 2.06 1.61	0.65–3.45 0.80–14.96 0.62–6.87 0.94–2.74			
Multifocality No Yes	Ref 0.36	1.60	0.58–4.41			
2004 biopsy grade LG HG	Ref 0.20	1.50	0.80–2.79			
Cytology Neg Atypical Positive for HG No Diag	Ref 0.11 0.16 0.74	2.01 1.89 0.85	0.85–4.77 0.76–4.68 0.32–2.24			
JJ preop No Yes	0.70	1.10	0.65–1.86			
URSS diag No Yes	Ref 0.35	0.78	0.47–1.31			
Distal ureter management - Extravesical - Transvesical - Endoscopic	Ref 0.44 0.37	0.79 1.51	0.43–1.44 0.60–3.83			
Distal margins Negative Positive	Ref 0.018 *	2.09	1.13–3.86	Ref 0.04 *	1.93	1.01–3.70

* <0.05.

**Table 6 cancers-14-05452-t006:** Cox regression for factors associated with HUR.

	Univariate			Multivariate		
Variable	*p*	HR	95% CI	*p*	HR	95% CI
Hydronephrosis No yes	Ref 0.72	1.17	0.47–2.90			
Size <2 cm >= 2 cm	Ref 0.02 *	3.33	1.13–9.76	0.02 *	3.59	1.22–10.60
cTN cTaN0 cT1N0 cT2N0 cTanyN1–2 cTxN0M0	Ref 0.87 0.99 0.41 0.03 *	0.90 NA 1.85 0.34	0.26–3.07 0.42–8.12 0.12–0.93			
Multifocality No Yes	Ref 0.07 *	3.04	0.91–10.2	0.04 *	3.44	1.01–11.71
2004 biopsy grade LG HG	Ref 0.24	1.63	0.71–3.70			
JJ preop No Yes	0.29	1.52	0.69–3.33			
URSS diag No Yes	Ref 0.25	1.66	0.69–3.97			
Past history NMIBC No Yes	Ref 0.36	1.42	0.66–3.04			
Synchronous NMIBC No Yes	Ref 0.37	1.47	0.62–3.49			
Distal margins Negative Postive	Ref 0.76	1.17	0.40–3.41			

* <0.05.

## Data Availability

Data supporting reported results can be found by writing to alexandra.massonlecomte@aphp.fr.

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
