# Peer review of "Oncological Outcomes of Distal Ureterectomy for High-Risk Urothelial Carcinoma: A Multicenter Study by The French Bladder Cancer Committee"

_cancers, 2022, doi:10.3390/cancers14215452_

Round 1

Reviewer 1 Report

Overall, this is a well written manuscript on an important management approach in patients with upper tract urothelial carcinoma. I think the study should be published with the following major revisions:

Pre-specified endpoint of p<0.05 is statistically significant. Please remove indication (*) for p<0.1 in the tables as this is not a meaningful indication. It is helpful to indicate variables with statistical significance but please remove the differentiation between p<0.05 and p<0.01. Both meet pre-specified level of significance.

In the text, p value is currently reported as p ### (for example, p 0.01). Please be more specific, if p=0.01, please add the appropriate indicator.

In Tables 1 and 2, please either report mean and standard deviation or median and interquartile range but not both. Please do not report min-max range at it does not tell meaningful information about your dataset.

Table 2, missing data under lymphadenectomy. Please provide more information in the manuscript regarding patients with Clavien 3- 5  complications (what surgeries did they receive, what were the complications)

In table 4, HG was not statistically significant on univariable analysis (p=0.07), please remove from multivariable analysis.

In table 6, tumor multifocality was not statistically significant on univariable analysis (p=0.07), please remove from multivariable analysis. Please revise statement from lines 239 – 243 regarding tumor multifocality was independent predictor of ipsilateral recurrence because it did not meet your prespecified statistical significance level on univariable analysis and should not be input in multivariable analysis. It does make sense intuitively and likely does in reality but your dataset did not support that. Likewise, please revise discussion section accordingly.

Reviewer 2 Report

This paper addresses a question commonly seen in practice. However, it is often a case by case decision on who gets spared nephrouretrectomy. The authors attempt here to report a historical cohort outcome to add to the existing renal preservation literature.
- It is unclear if documentation of a disease-free renal pelvis was done in all patients? I.e. what was the percentage of negative washing cytology in renal pelvis before sparing the kidney? What percentage was not available? was this repeated during resection surgery? 
- numbers in table 1 under the breakdown of grade don’t add up to %100 between low grade, high grade and unknown. 
- once the numbers are rectified, the high grade patients deserve a dedicated analysis since they are the ones controversial for renal preservation.

- discuss how to integrate this into the existing renal preservation literature. This is particularly interesting in patients with existing CKD, solitary kidney etc (see PMID 35475152).

Round 2

Reviewer 1 Report

Appropriate revisions compared to first version. Need to spell check and fix some typos prior to final publication but authors can work with copy editor for that.